# Impact of Binge Alcohol Intoxication on the Humoral Immune Response during *Burkholderia* spp. Infections

**DOI:** 10.3390/microorganisms7050125

**Published:** 2019-05-09

**Authors:** Ryan M. Moreno, Victor Jimenez, Fernando P. Monroy

**Affiliations:** 1Department of Biological Sciences, Northern Arizona University, Flagstaff, AZ 86011, USA; rmm326@nau.edu (R.M.M.); vj92@nau.edu (V.J.J.); 2Pathogen & Microbiome Institute (PMI), Northern Arizona University, Flagstaff, AZ 86011, USA

**Keywords:** *Burkholderia*, melioidosis, alcohol, adaptive immune system, antibodies, immunoglobulins, binge alcohol, humoral, virulence

## Abstract

*Burkholderia pseudomallei*, the causative agent of melioidosis can occur in healthy humans, yet binge alcohol use is progressively being recognized as a major risk factor. Currently, no experimental studies have investigated the effects of binge alcohol on the adaptive immune system during an active infection. In this study, we used *B. thailandensis* and *B. vietnamiensis,* to investigate the impact of a single binge alcohol episode on the humoral response during infection. Eight-week-old female C57BL/6 mice were administered alcohol comparable to human binge drinking (4.4 g/kg) or PBS intraperitoneally 30 min before intranasal infection. Mice infected with *B. thailandensis* had a 100% survival rate, while those infected with *B. vietnamiensis* had a 33% survivability rate when a binge alcohol dose was administered. *B. thailandensis* was detected in blood of mice administered alcohol at only 7 days post infection (PI), while those infected with *B. vietnamiensis* and receiving alcohol were found throughout the 28-day infection as well as in tissues at day 28 PI. Binge alcohol elevated IgM and delayed IgG specific to the whole cell lysate (WCL) of *B. vietnamiensis* but not *B. thailandensis* infections. Differences in immunogenicity of *B. pseudomallei* near-neighbors provide a framework for novel insights into the effects of binge alcohol’s suppression of the humoral immune response that can cause opportunistic infections in otherwise healthy hosts.

## 1. Introduction

*Burkholderia pseudomallei* (*B. pseudomallei*), a gram-negative soil-borne bacterium, is the causative agent of melioidosis [1], a disease endemic in Northern Australia and Southeast Asia with an increasing presence in tropical regions globally [2]. *B. pseudomallei* has numerous virulence factors that allow it to survive and thrive in varying environments. Melioidosis is generally characterized as a severe pneumonic infection, by its ability to colonize and infect phagocytic cells within the lungs. This disease causes 54% mortality globally due to difficulties in identifying disease symptoms and the high antibiotic resistance of this organism to commonly prescribe antibiotics [1]. As an emerging tropical disease, melioidosis research has increased to fully understand the epidemiology, disease dynamics, and risk factors involved [3].

Increasing epidemiological evidence indicates that specific risk factors for melioidosis may be of greater influence than infectious dose, route of infection, or bacterial virulence for developing melioidosis [2]. A major risk factor for melioidosis is binge alcohol intoxication, present in 39% of cases [4,5,6]. Unlike chronic alcoholism or more severe alcohol use disorders (AUDs), studies in both human and animal models indicate that binge alcohol intoxication is a risky pattern characterized by the consumption of 4–6 standard drinks or reaching a minimum blood alcohol concentration (BAC) of 0.08% or higher within a 2–3-h drinking episode [7]. Moreover, the effects of chronic alcoholism on lung immunity are well documented; however, the modulating effects of binge alcohol intoxication on the humoral immune response during infection and in particular, *Burkholderia* infection have not been described [8,9]. In our previous studies, we found binge alcohol conditions alter alveolar macrophage phagocytosis, reactive nitric oxide (RNS) production, and increased intracellular survival of *Burkholderia thailandensis* in vitro [10]. Similarly, the effects of alcohol during infection are well documented [6,11]. Not only does alcohol have an effect on the innate immune system, but it also has impactful effects on the adaptive immune system [12]. Chronic alcoholics are often lymphopenic and have a decreased response to mitogen stimulation [13]. Patients diagnosed with alcohol use disorder, express significantly lower amounts of immunoglobulin levels in the lungs compared to their healthy counterparts [7].

The loss of peripheral B cells associated with alcohol abuse seems to be, in part, due to a decrease in the frequency of B-2 B cells [14]. Moreover, adaptive immunity mediated by B-2 B cells relies on the specific role of antibodies upon activation and differentiation. Conventional B-2 B cells upon activation produce high affinity antibodies, develop into long-lived memory B cells, and undergo class switching to aid in the removal of the pathogen. B-1 B cells is a second B cell population that are considered part of the innate immune system for their ability to self-renew, recognize of T cell independent antigens, and produce IgM and IgA upon stimulation [15]. The effects of binge alcohol intoxication on the effector functions of B cells during *B. pseudomallei* near-neighbor infection has not been investigated, and the basis for our working hypothesis. Our research sought to investigate the effects of binge alcohol intoxication, utilizing a murine model and two different *Burkholderia* species: *B. thailandensis* E264, a genetically similar strain to *B. pseudomallei* [16] and *B. vietnamiensis*, a known human pathogen found in the *B. cepacia* complex that commonly infects cystic fibrosis patients [17]. By using both of these bacteria, we were able to show different perspectives as to how binge alcohol intoxication impacts the B-cell mediated humoral immune responses and allows less-pathogenic *Burkholderia* spp. the ability to persist and cause infection in vivo. Furthermore, our study investigated the effects of binge alcohol on IgM and IgG responses to major *Burkholderia* antigens important in bacterial clearance. Our results indicated that there was a delay and significant suppression in IgG in the infected and alcohol treated mice compared to that of infected but non-alcohol treated.

## 2. Materials and Methods

### 2.1. Animals for Study

The protocols used to determine the effects of binge alcohol consumption on the humoral immune response in mice were approved by Northern Arizona University Institutional Animal Care and Use Committee (IACUC) in accordance with federal regulations regarding the use of animals in research, (approval number 16-006). The binge alcohol intoxication mouse models used were 8-week old female C57BL/6J mice (Bar Harbor, ME, USA).

### 2.2. Bacterial Preparation

For each study, frozen stock cultures (*B. thailandensis* E264 or *B. vietnamiensis* Florida, USA strain) were inoculated into Luria Bertani broth (LB) and incubated overnight at 37 °C in an orbital shaker incubator (200 rpm) (New Brunswick C25, Edison, NJ, USA). Bacteria were diluted 1:10 and grown to late-logarithmic phase measured by optical density at OD_600_ absorbance in a spectrophotometer (Eppendorf Bio Photometer AG2233, Hamburg, Germany). Bacteria were collected in 1 mL by centrifugation and resuspended in 1 mL with pre-warmed Dulbecco’s Phosphate-Buffered Saline (PBS) at a concentration of 1 X 10^5^ cfu/25μL. Actual numbers of viable bacteria were determined by standard plate counts of the bacterial suspensions on LB agar plates. The Pathogen & Microbiome Institute (PMI), Northern Arizona University, Flagstaff, USA, kindly provided the *Burkholderia* strains.

### 2.3. Experimental Design, Alcohol Administration, and Blood Collection

Animals were housed in separate cages based on the specific treatment group throughout each experiment and were given food and water ad libitum. Mice were separated into 4 groups: Mice not administered alcohol and not infected (CON; *n* = 10); mice administered alcohol and not infected (ALC; *n* = 10); mice not administered alcohol and infected (INF; *n* = 10); and mice administered alcohol and infected (ALC+INF; *n* = 10). No statistical differences were found between CON or ALC groups in any assay. Alcohol was prepared from a 100% stock solution and was diluted to 20% *v*/*v* in sterile PBS. On the day of experiments, mice were administered (4.4 g/kg of 20% *v*/*v* alcohol) or PBS into the peritoneum. After 30 min post alcohol administration, blood alcohol concentrations (BAC) reached 0.254% and infected groups were inoculated intranasally with 25 μL of media containing 1 × 10^5^ colony-forming units (CFUs) of *B. vietnamiensis* or *B. thailandensis,* or PBS under light anesthesia. This BAC represents the higher end of the range observed in humans, but it is not particularly uncommon and has been reported as a common range in human binge drinkers in a number of studies [18]. Equally, mice eliminate alcohol from their systems more rapidly than humans. Producing biologically equivalent effects of alcohol in mice, as in human binge drinkers, requires a higher dosage in mice. Blood alcohol concentration measurements and quantification of bacteria were conducted as described in Jimenez et al. [18]. Actual numbers of viable bacteria were determined by standard plate counts using limiting dilution analysis of the bacterial suspensions on LB agar plates.

Steps were taken to prevent cross-contamination such that, handling and phlebotomy were performed on non-infected groups first and mice that were infected were separated amongst the ventilation racks. Every 7 days throughout the 28-day experiment, a volume of blood equal to 1% of each mouse’s body weight was collected via the lateral tail vein on days 7, 14, and 21. On day 28 PI, mice were euthanized by CO_2_ asphyxiation and blood collected by a terminal cardiac bleed. Mice were observed daily for clinical signs and symptoms of infection. At least two independent animal experiments were run with similar results.

### 2.4. Lipopolysaccharide (LPS), Capsular Polysaccharide (CPS), and Whole Cell Lysate (WCL) Preparation

The *B. pseudomallei* CPS and LPS antigens were kindly provided by Dr. Paul Brett and Dr. Paul Keim, respectively, and their preparation has been previously described [19,20]. WCL was prepared from an overnight culture of a single CFU from both a *B. thailandensis* and *B. vietnamiensis* culture plate. Culture tubes were pelleted by centrifugation at 13,000 rpm for 10 min. The bacterial pellet was resuspended in lysis buffer (50 mM Tris, 150 mM NaCl, 1% Triton X-100, 10 mM EDTA, pH 7.2) and subjected to 3 freeze-thaw cycles, which each consisted of 30 min on dry ice and 30 min at room temperature. Samples were sonicated (S-3000, Misionix, Farmingdale, NY, USA) for 3 min to ensure complete lysis. Protein concentration in the lysate was determined using the PierceTM BCA Protein Assay Kit following manufacture’s protocol (Thermo Scientific, Rockford, IL, USA) and absorbance determined in a Synergy HT Plate Reader (BioTek, Winooski, VT, USA) at 450 nm. Protein determination assay was repeated independently at least twice with similar results.

### 2.5. Enzyme-Linked Immunosorbent Assay

Prior to determining total IgM and IgM antibody levels and specific responses to whole cell lysate (WCL) in mouse serum, the optimal conditions for the ELISA were determined. Total IgM and IgG concentrations were determined using a Mouse IgM and IgG total Ready-SET-Go! ELISA Kit (Affymetrix eBioscience, Vienna, Austria) following the manufacturer’s protocol. The optimal dilution of serum samples was 1:20,000 calculated from the linear portion of the assay’s standard curve, established with known concentrations of IgG and IgM. A 1:2000 dilution for IgM and 1:200 dilution for IgG were used to determine specific responses to the WCL. Costar 96-well plates coated with WCL at 2.5 µg/mL by overnight incubation at 4 °C. The secondary antibodies, HRP conjugated rabbit anti-mouse IgM or IgG (DAKO, Copenhagen, Denmark), were used at dilutions of 1:500 and 1:2000, respectively. ELISAs were developed using TMB substrate (Invitrogen, Camarillo, CA, USA). Results were determined as absorbance value (OD 405 nm) with a Synergy HT Plate Reader and the average OD values of triplicate wells were used for analysis. While titers could not be determined using this approach, we could still make comparisons in the antibody responses between the infected groups. ELISA assays were repeated independently at least twice with similar results.

### 2.6. MagPix Detection Assays: Coupling of Antigen to Beads

*Burkholderia* antigens, capsular polysaccharide (CPS) or lipopolysaccharide-A (LPS) (kindly provided by Dr. Paul Brett and Dr. P. Keim, respectively) were coated to beads as described by Schlottmann et al. [21]. *Burkholderia* whole cell lysate (WCL) were covalently coupled to carboxylated magnetic Luminex microspheres through a carbodiimide reaction with the use of an xMAP^®^ Antibody Coupling Kit (Luminex Corp, Austin, USA). Manufacturer’s instructions, as previously detailed by Perraut et al. [22], were followed. Briefly, 2.5 × 10^6^ beads from regions 20 to 40 were used in a working volume of 500 µL. After carbodiimide hypochloride (EDC) activation step, 5 µg of antigen per million beads was added in the activation buffer and kept under rotation mixing in the dark for 2 h. After pelleting and washing, the supernatant was removed and replaced by 1 mL wash buffer and kept in the dark at 2–8 °C. Final count of remaining beads using cell counter showed a mean recovery of 98% of the coupled beads. Efficient coupling of antigen was controlled using positive monoclonal antibodies to both antigens. We achieved close to 90% coupling efficiency. The coupled microspheres were kept in the washing/storage buffer at 4 °C in the dark until use.

### 2.7. Bead-Based Assay for IgM and IgG Antibodies

Coated microspheres were added to a white, polystyrene, opaque, round-bottomed microtiter plate (Fisher Scientific, Illkirch, France). Plates included one positive control: A pool of mouse sera previously infected with *B. thailandensis* and *B. vietnamiensis*. Serum pools from non-infected mice were included as negative control. Briefly, 2.5 µL aliquots containing 2000 antigen-coated beads were dispensed to individual wells together with 100 μL of serum diluted 1:100 with PBS Tween 0.01% BSA 1% (PBSB). A microplate shaker (IKA^®^MTS, Wilmington, NC, USA) was used to mix the suspension at 350 rpm for 45 min, while shielded from light. Following two washes with 100 μL PBSB, 100 μL of goat anti-mouse biotinylated antibody (4 µg/mL) was added and suspensions shaken for 45 min. After removing supernatants, a 1:1000 dilution of streptavidin R-phycoerythrin conjugate (Life Technologies) was added to all wells. Plates were incubated at room temperature for 1 h with shaking. After two washes with 100 μL/well of PBSB, the beads were then resuspended in 120 μL PBSB and analyzed on a Multiplex MAGPIX system (Millipore, Burlington, MA, USA) using the xPONENT 4.1 manufacturer’s software for acquisition. Antibody binding was represented by median fluorescence intensity (MFI). For a result to be valid, a minimum of 50 beads/well must be detected by the software. MagPix assays were run in triplicate and provided validation for ELISA assays.

### 2.8. Data Analysis

The statistical analysis was performed using Prism 7.0 software (Graph Pad, 7.0, San Diego, CA, USA). Antibodies in each treatment group were tested for normal variance and a normal distribution using a Shapiro-Wilk’s Test for normality. Treatment groups were then analyzed using two-way ANOVA with Bonferroni adjustment for multiple comparisons. *P*-values of less than 0.05 were considered statistically significant.

## 3. Results

### 3.1. Effect of Binge Alcohol Intoxication on Mouse Survival Following Infection with Burkholderia Species

To investigate the effects of a single binge alcohol episode on morbidity or mortality during a pneumonic infection with less pathogenic *Burkholderia* species, the survival of mice was closely analyzed for 28 days post infection (PI). By the conclusion of the experiments, mice infected with *B. thailandensis* had a 100% survival rate, while mice infected with *B. vietnamiensis* had a 33% survivability rate in the binge alcohol group (Figure 1). Most mortality occurred on day 24 PI. There was no mortality in the non-infected control groups, or in mice only administered alcohol. Weight loss and lethargy were observed in ALC+INF mice but it was not significant and returned to start weight in week 2. Figure 2A shows CFU in blood collected every 7 days PI. Only at day 7 PI in the ALC+INF mice with *B. thailandensis* were bacteria detected, while those infected with *B. vietnamiensis* were detected in both the INF and the ALC+INF mice. After day 14 PI only the ALC+INF mice with *B. vietnamiensis* had detectable bacteria in the blood until day 28 PI. When tissues were collected and analyzed at day 28 PI, we found presence of bacteria (CFU) in all tissues analyzed (Figure 2B)

### 3.2. Binge Alcohol Selectively Suppresses Total IgM Concentration in B. thailandensis but Not in B. vietnamiensis Infections

Total IgM concentrations can be an indicator of the initial humoral immune response against infection. Testing binge alcohol related immune dysfunction required phlebotomy from each mouse weekly for 28 days. Within three weeks PI, mice in the INF group with *B. thailandensis* expressed IgM levels similar or lower than those in control mice (CON, ALC). By day 28 PI IgM levels in the ALC+INF group were reduced two-fold compared to control groups. Furthermore, total IgM in ALC+INF mice with *B. thailandensis* was significantly decreased throughout the 28-day study when compared to control INF mice (Figure 3A). When using the more virulent *B. vietnamiensis* strain, different IgM responses were observed compared to *B. thailandensis* infected humoral responses. Mice in the ALC+INF group expressed total IgM comparable to CON, ALC, or INF groups until day 14 PI. By day 21 PI, total IgM concentrations in the ALC+INF mice were significantly elevated compared to all other groups. Total IgM concentration in ALC+INF mice remained elevated by day 28 PI but there were no differences when compared to INF mice (Figure 3B). Taken together, this data indicates that a single binge alcohol episode can selectively delay the rate of the IgM humoral response depending on the type of *Burkholderia* spp., infection.

### 3.3. Binge Alcohol Delays Total IgG Concentrations during Burkholderia Infection

IgG antibodies are generated following the initial IgM response and this class of antibody is commonly associated with control and resolution of infection. Mice in the ALC+INF group infected with *B. thailandensis* expressed similar total IgG concentrations to INF mice until day 21 PI. A significant decrease in total IgG concentration between the ALC+INF mice compared to INF mice was measured at day 14 PI. At day 28 PI levels of total IgG increased in both the INF and ALC+INF mice with significant higher levels found in the ALC+INF mice (Figure 4A). Conversely, ALC+INF mice infected with *B. vietnamiensis* expressed a decrease in total IgG levels throughout the 28-day experiment. Significant differences were found at day 14 PI between the INF and ALC+INF mice (Figure 4B). This data indicates that binge alcohol can suppress or delay the IgG antibody responses depending on the type of *Burkholderia* spp. infection.

### 3.4. Binge Alcohol Elevates IgM and Delays IgG Specific to the Whole Cell Lysate (WCL) of B. vietnamiensis but Not B. thailandensis Infections

Specific antibodies to WCL are good indicators of the host immune response to the pathogen. Antibodies specific to each *Burkholderia* species were quantified in the serum of infected mice. Non-infected mice were not utilized to examine specific antibody response to infection. In ALC+INF mice with *B. thailandensis* expressed lower specific IgM on day 14 PI, followed by significantly higher concentrations at day 21 and 28 PI. Specific IgM in ALC+INF mice remained elevated above INF mice at day 28 PI (Figure 5A). When measuring specific IgG for *B. thailandensis* WCL, IgG concentrations were significantly decreased in ALC+INF mice throughout the 28-day experiment compared to INF mice but significant differences were only found at day 21 PI (Figure 5C). Interestingly, IgM and IgG specific to *B. vietnamiensis* WCL antigens elicited a greater humoral immune response compared to mice infected with *B. thailandensis*. In ALC+INF mice a 3-fold increase in IgM levels were observed when compared to INF mice at days 14, 21, and 28 PI (Figure 5B). Conversely, specific IgG concentrations were suppressed ~threefold in ALC+INF mice compared to INF mice at days 7, 14, and 21 PI. At day 28 PI, a significant decreased was still observed in the ALC+INF mice (Figure 5D). The data indicates that a single binge alcohol intoxication bout can selectively alter the specific humoral immune response to a particular pathogen. While levels of IgM were usually increased those of IgG were decreased regardless of the pathogen.

### 3.5. Binge Alcohol Suppresses IgM and IgG Specific to the Capsular Polysaccharide (CPS) of B. vietnamiensis

Humoral responses against the capsular polysaccharide of *Burkholderia* species have been proposed to be protective and a good target for vaccination [17]. To further investigate the effects of alcohol on concentrations of specific antibodies produced in WCL against *B. vietnamiensis*, we sought to find the proportion of IgM and IgG specific to CPS. The CPS used was from *B. pseudomallei*. We did not analyze CPS-specific antibody responses in *B. thailandensis* because the strain used, *E264* does not express a CPS-like structure [23]. In ALC+INF mice a significantly decrease was observed in specific IgM antibodies to CPS at days 14 and 28 PI. While not statistically significant, less IgM was also measured in ALC+INF mice at day 21 PI (Figure 6A). When analyzing IgG specific to CPS, a similar trend was observed in the ALC+INF mice. At days 21 and 28 PI, ALC+INF mice infected and expressed significantly less IgG antibody compared to INF mice (Figure 6B). These findings suggest that binge alcohol intoxication decrease both IgG and IgM humoral antibody response that preferentially targets polysaccharide motifs.

### 3.6. Binge Alcohol Suppresses IgM Specific to the Lipopolysaccharide-A (LPS-A) of B. thailandensis and B. vietnamiensis, While IgG Remained Unchanged

IgM and IgG antibodies that specifically target the LPS of bacteria facilitate a more specific immune response for pathogen clearance. The effects of alcohol on an additional immunogenic component of *Burkholderia* spp. were analyzed. In ALC+INF mice with *B. thailandensis* we saw a significantly decrease in IgM specific to LPS at days 14 and 21 PI, compared to INF mice. IgM concentrations were comparable at day 28 PI in both groups (Figure 7A). When evaluating IgG specific to *B. thailandensis* LPS-A, ALC+INF mice expressed a ~twofold increase in IgG compared to INF mice at day 7 PI. No statistical differences were found between groups at any other time point (Figure 7C). The suppressive effect was more pronounced in the ALC+INF mice with *B. vietnamiensis* were a significantly decrease in IgM specific to LPS was observed at days 14, 21, and 28 PI compared to INF mice. The peak IgM concentration in the INF mice was reached at day 14, while in the ALC+INF group the peak was reached at day 21 PI (Figure 7B). While lower concentrations in IgG specific for LPS-A were measured in mice ALC+INF mice at days 14 and 28 PI, there were no significant differences between treatment groups throughout the 28-day experiment (Figure 7D). Our data support the fact that binge alcohol suppressed antibody responses but preferentially targeted the IgM response against both pathogens.

## 4. Discussion

Melioidosis affects 165,000 humans every year, with the majority of cases originating in endemic areas, Thailand, Malaysia, Singapore, and Northern Australia [3]. Infection with *B. pseudomallei* causes severe pneumonia and more rarely, septicemia and meningitis. Different risk factors contributing to melioidosis morbidity includes diabetes and alcohol abuse [24]. It has been reported that ~39% of patients that are *B. pseudomallei* culture positive report a recent binge alcohol intoxication episode [2]. The effects of binge alcohol on the *Burkholderia*–host immune system interaction remain unclear.

Studies in patients and mouse models have provided evidence that both humoral and cellular immunity are needed to confer protection. The intracellular nature of *B. pseudomallei* and its ability to survive and multiply in the intracellular environment establish the need to generate cell-mediated immune responses to combat infection. Cellular responses seem to be initially related to protection against disease progression and later clearance of infection in which both CD8+ and CD4+ T cells play a crucial role by producing IFN-γ [25]. In addition, protection against infection by *B. pseudomallei* correlates well with the elicitation of a secondary systemic and mucosal antibody response. High levels of IgG, IgA and IgM are found in serum from patients with melioidosis, with higher titers found in patients with invasive diseases than localized diseases [26].

In the present study, we studied the effects of a single binge alcohol episode on the adaptive immune system during infection with a less-pathogenic (*B. thailandensis*) and an opportunistic, *B. vietnamiensis* near neighbor of *B. pseudomallei* [16]. Our results supported our initial hypothesis that binge alcohol intoxication resulted in a prolonged infection due in part to a dysfunction in the host humoral response that may aid in the survival of less-pathogenic and opportunistic pathogens. In our study, 67% of ALC+INF mice with *B. vietnamiensis* died by 28 days PI. Whereas no mortality occurred in mice not administered alcohol prior to infection, including those infected with *B. thailandensis.* These findings were not surprising since *B. vietnamiensis* is an opportunistic pathogen in immunocompromised patients such as those suffering from cystic fibrosis [27]. Furthermore, *B. vietnamiensis* also expresses more virulence factors than *B. thailandensis* [28]. Species within the *B. cepacia* complex produce exopolysaccharides (EPS) polymers [29]. The EPS found in *B. vietnamiensis* could be important in evading phagocytosis and persist in circulation in the presence of alcohol. In addition, this was an environmental isolate recently collected from Florida [30]. To our knowledge this is the first time this isolate has been used to experimentally infect a host. This may in part explain some of the observed virulence.

Furthermore, *B. thailandensis* has been shown in culture to produce more biofilms in the presence of alcohol compared to control groups [9]. *B. pseudomallei* near-neighbors ability to produce more biofilm in the presence of alcohol could partially explain why there was an increase in mortality within the ALC+INF mice compared to the other treatment groups (Figure 1). Interestingly, while mice infected with *B. thailandensis* had controlled the infection by day 7 PI, mice in the ALC+INF group were unable to do so and may have been in part to a dysfunctional phagocytic system that we have observed in vitro when alcohol is present [9]. In the ALC+INF mice infected with *B. vietnamiensis,* bacteria were found in the blood until day 28 post-infection. While bacteria levels (CFU) were low in the blood of these animals, CFU were elevated in several tissue suggesting intracellular survival and colonization. Whether this bacteremia caused the observed mortality is not known but future studies will focus on collecting tissues in the ALC+INF mice at days 14 and 21 PI to determine if lower bacterial numbers in the blood are associated with higher tissue colonization.

Three antigens were selected (WCL, CPS, and LPS) to further analyze the effects of binge alcohol on the specific IgG and IgM responses during a *Burkholderia* infection. These antigens are important because of their application in diagnosis, protection, and vaccine development [31,32,33,34,35]. Humoral responses under alcoholic conditions were generally suppressed for both IgM and IgG despite the fact that bacteria had already been eliminated from the host (*B. thailandensis*) or still present in host tissues (*B. vietnamiensis*). Interestingly, the complement system and innate immunity may have significant effects in bacterial clearance among acapsular *B. thailandensis* and capsular producing *B. vietnamiensis*. It was expected that polysaccharides present in LPS-A and CPS would preferentially induce strong IgM responses which is typically associated with T-cell-independent B cell responses [36]. Interestingly, the monoclonal antibody 3C5 used in the commercial lateral flow assay to identify CPS from *B. pseudomallei*, does not recognize CPS from *B. vietnamiensis* [32]. Regardless, we have indirect evidence that the CPS from *B. pseudomallei* cross-reacts with CPS from *B. vietnamiensis* as shown by the increased absorbance values in both the ELISA and MagPix assays.

Moreover, it was not the intent of the current study to identify immunogenic antigens, but rather evaluate the overall humoral response to a *Burkholderia*-type infection in the context of a single bout of binge alcohol, future studies should include further analysis of host antibody responses to intact (ELISA) and denatured (Western blot) *Burkholderia* antigens. Alcohol decreases the numbers of peripheral B cells, in particular B-2 B cells, the B cell population producing high affinity antibodies due to antibody class switching and can provide memory upon activation and clonal expansion [13,34]. Future studies with our mouse model of binge alcohol and infection will focus on the role of serum cytokine levels and the characterization of B cell populations throughout the infection. It is possible to speculate that alcohol drives B-1 B cell populations during *Burkholderia* spp. infection leading to T-cell independent responses and mostly broad reactive IgM and IgA antibodies. Furthermore, B-1 B cells are known to recognize capsular polysaccharides [37] and react with oxidized phospholipids and reactive oxygen species produced during alcohol metabolism, leading to a preferential production of IgG3 antibodies [37].

The effects of alcohol on humoral immunity during infection have shown that alcohol interferes with B-lymphocyte development and increases the risk of infections such as pneumonia, HIV, and tuberculosis [8,35]. In a separate study it was found that alcohol decreased both B-lymphocyte numbers in the upper and lower airways, as well as a decrease in the levels of protective IgG. These results allowed for increased incidence of infection in the lungs and further showed a dysfunction in adaptive immunity with the addition of hazardous alcohol consumption [13]. While it is premature to assume that a single episode of binge alcohol is involved in the impairment of IgM to IgG class switching, it is clear that alcohol has a direct effect on serum cytokine production. For example, binge alcohol affects cytokine production by human monocytes leading to increased Transforming Growth Factor (TGF)-β1 [12], and decreased IL-4 activation on murine B-lymphocytes [38]. These cytokines are essential for B cell to effectively undergo class switch from IgM to IgA and IgG1, respectively [39]. Further experiments must be performed to evaluate blood cytokine levels during infection to gain a better understanding on the effects of a single binge alcohol episode in altering host humoral responses.

Moreover, our results indicate that a single binge alcohol episode can lead to a dysfunction in the adaptive immune system, specifically the humoral immune response. Future studies will focus on using more sensitive methods (i.e., PCR) to document infection in blood and tissues. In addition, tissues could be used in histopathology to determine if alcohol not only contributes to bacteria burden but also to tissue pathology. Differences in immunogenicity of *B. thailandensis* and *B. vietnamiensis* provide a framework for novel insights into the effects of binge alcohols’ suppression of IgM antibodies and irregular IgM production in relation to IgG antibodies, respectively. It is clear that in the presence of alcohol *B. pseudomallei* near neighbors can become opportunistic pathogens in otherwise healthy hosts. We can speculate that binge alcohol-like conditions would improve *B. pseudomallei* survival and immuno-modulating factors during infection, such as type-three secretion systems (TTSS) or cell-surface polysaccharide mediated immunological avoidance (i.e., CPS, EPS) that are conserved across *B. pseudomallei* and *B. thailandensis* or *B. vietnamiensis*, respectively. This mouse data supports the findings that binge alcohol intoxication is a predisposing factor for melioidosis in otherwise healthy hosts. Future research will investigate serum cytokine production, B cell population targeted by alcohol, and the mechanisms associated in humoral immune dysfunction to permit bacterial dissemination during binge alcohol episodes. Most importantly, how an increase of blood alcohol levels affects the high pathogenic *B. pseudomallei* and melioidosis rates of infection.

## Figures and Tables

**Figure 1 microorganisms-07-00125-f001:**
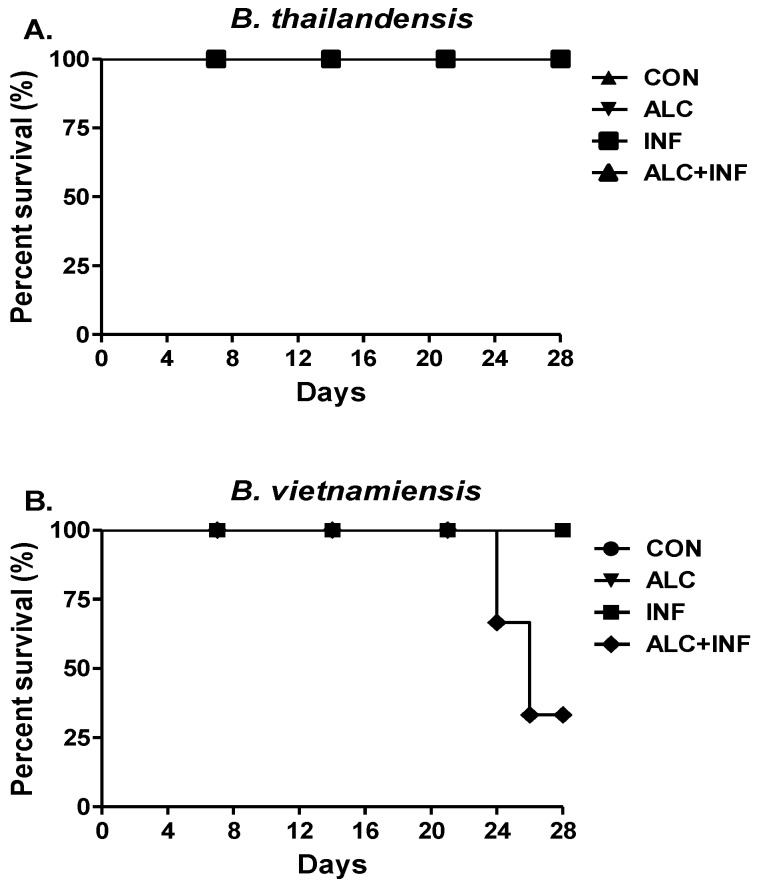
The effects of binge alcohol intoxication on mice survival during *B. pseudomallei* near-neighbor infections. (**A**) Percent survival of mice after *B. thailandensis* and (**B**) *B. vietnamiensis* infection. Mice were injected intraperitonially with 20% *v*/*v* of alcohol 30 min prior to intranasal infection with 1 × 10^5^ CFU for both bacterial strains. Survival was monitored for 28 days. Control group (CON) was not administered alcohol or infected.

**Figure 2 microorganisms-07-00125-f002:**
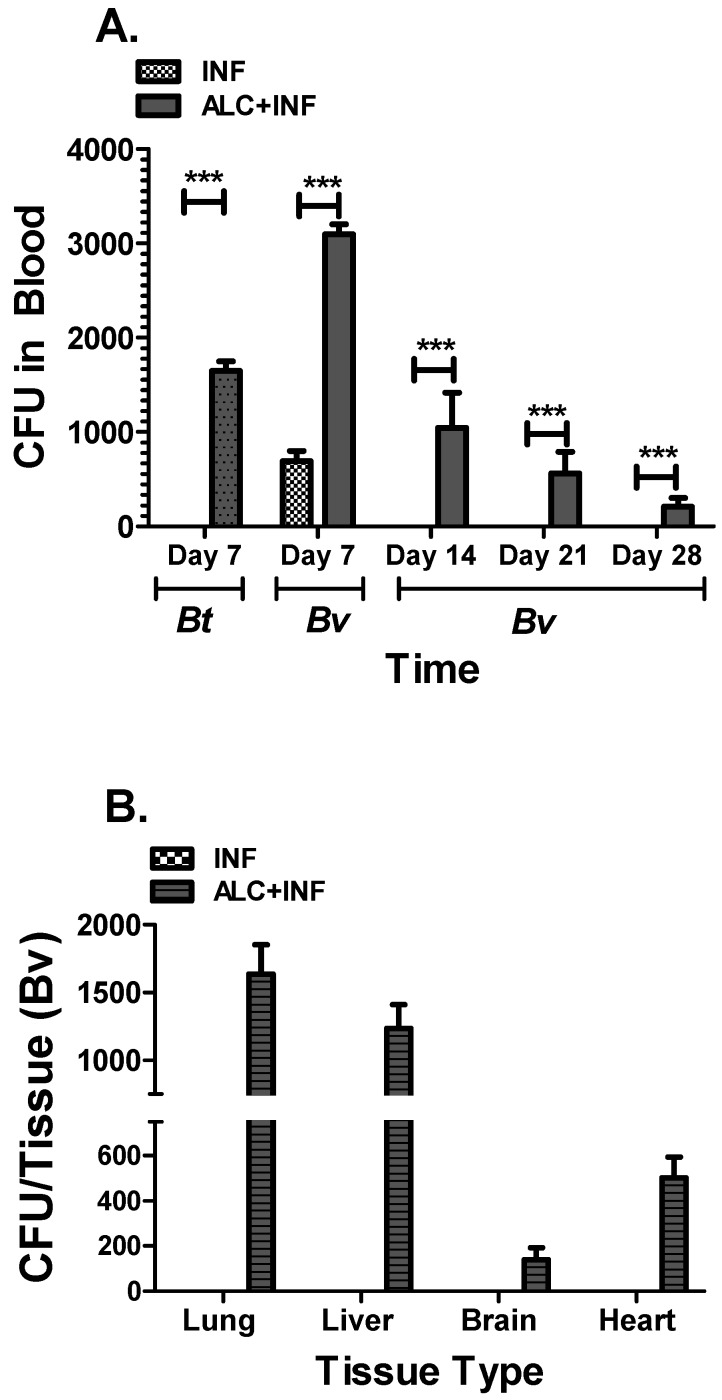
Whole blood and tissue colony forming units (CFUs) from mice infected with *B. thailandensis* (*Bt*) or *Burkholderia vietnamiensis* (*Bv*). (**A**) Blood was collected weekly post infection and *Burkholderia* species were grown on LB media plates for 48 h at 37 °C to determine CFUs. (**B**) Tissues were collected and homogenized at day 28 from Bv-infected mice (INF) and mice infected with Bv and exposed to alcohol (ALC+INF) to determine bacterial tissue burden. Bars represent the average CFUs per treatment with SEM. Horizontal line with asterisk (*) represents statistical comparison to the INF group determined by Student’s *t*-test, *n* = 3. ** *p* ≤ 0.01, *** *p* ≤ 0.001.

**Figure 3 microorganisms-07-00125-f003:**
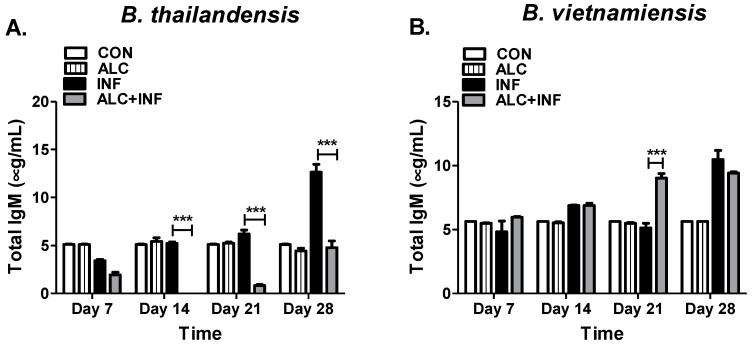
Total IgM concentration in mice infected with *B. thailandensis* or *B. vietnamiensis*. Mice were administered alcohol (4.4 g/kg) or PBS (i.p). At 30 min post binge alcohol, mice were intranasally infected with (**A**) *B. thailandensis* or (**B**) *B. vietnamiensis* (1 × 10^5^). Whole blood was collected weekly until 28 days PI. Total IgM in mice were determined by ELISA with an optimal 1:20,000 dilution. Horizontal line with asterisk (*) represents statistical comparison between INF and ALC+INF groups among weekly blood collections determined by one-way ANOVA, *n* = 3. *** *p* ≤ 0.001.

**Figure 4 microorganisms-07-00125-f004:**
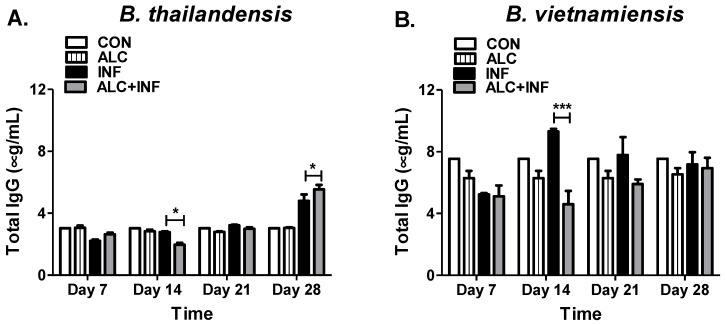
Total IgG concentration in mice infected with *B. thailandensis* or *B. vietnamiensis*. Mice were treated as described in Figure 3 and intranasally infected with (**A**) *B. thailandensis* or (**B**) *B. vietnamiensis* (1 × 10^5^). Whole blood was collected weekly until 28 days PI. Total IgG in mice were determined by ELISA with an optimal 1:20,000 dilution. Horizontal line with asterisk (*) represents statistical comparison between INF and ALC+INF groups among weekly blood collections determined by one-way ANOVA, *n* = 3. * *p* ≤ 0.05, *** *p* ≤ 0.001.

**Figure 5 microorganisms-07-00125-f005:**
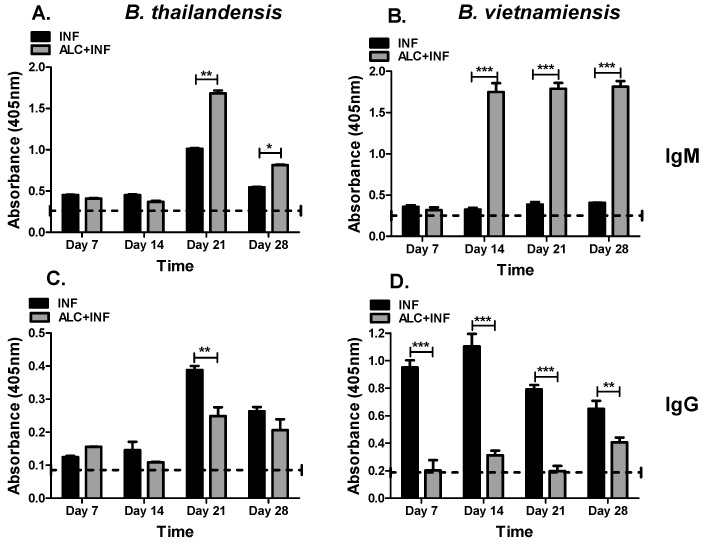
Specific IgM and IgG for *B. thailandensis* and *B. vietnamiensis* whole cell lysate (WCL). The mice were treated as described in Figure 3. Whole blood was collected weekly until 28 days PI. WCL was prepared from overnight cultures and 5 μg/mL were used to coat 96-well plates. Serum samples were used at a 1:2000 for IgM and 1:200 dilution for IgG. (**A**) Concentration of IgM specific to *B. thailandensis* and (**B**) *B. vietnamiensis* WCL. (**C**) Concentrations of IgG specific to the WCL of *B. thailandensis* and (**D**) *B. vietnamiensis* WCL. Dashed lines represent base values of control groups (CON or ALC). Bars represent average absorbance (405 nm) per treatment with SEM. Horizontal line represents statistical comparison between INF and ALC+INF among weekly blood collections determined by Student’s *t*-test, *n* = 3. * *p* < 0.05, ** *p* ≤ 0.01, and *** *p* ≤ 0.001.

**Figure 6 microorganisms-07-00125-f006:**
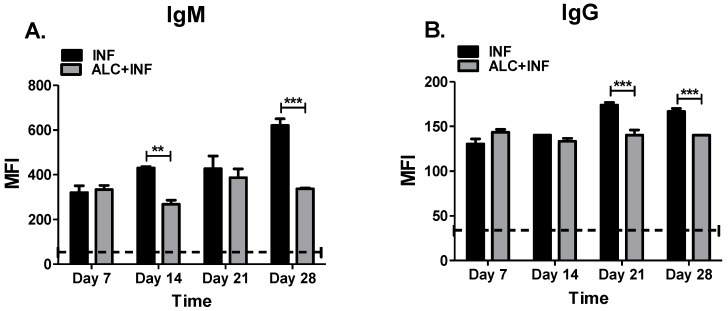
Specific IgM and IgG for capsular polysaccharide (CPS). The mice were treated as described in Figure 3. Whole blood was collected weekly until 28 days PI. *B. pseudomallei* CPS antigen (5 μg/mL) was coupled to MagPix microspheres and reacted with a 1:1000 serum dilution from infected mice (INF) or mice infected and exposed to alcohol (ALC+INF). Specific CPS IgM (**A**) and IgG (**B**) were measured from serum samples. Bars represent mean fluorescence intensity (MFI) per treatment with SEM. Dashed lines represent base values of control groups (CON or ALC). Horizontal line represents statistical comparison between INF and ALC+INF among weekly blood collections determined by Student’s *t*-test, *n* = 3. ** *p* ≤ 0.01 and *** *p* ≤ 0.001.

**Figure 7 microorganisms-07-00125-f007:**
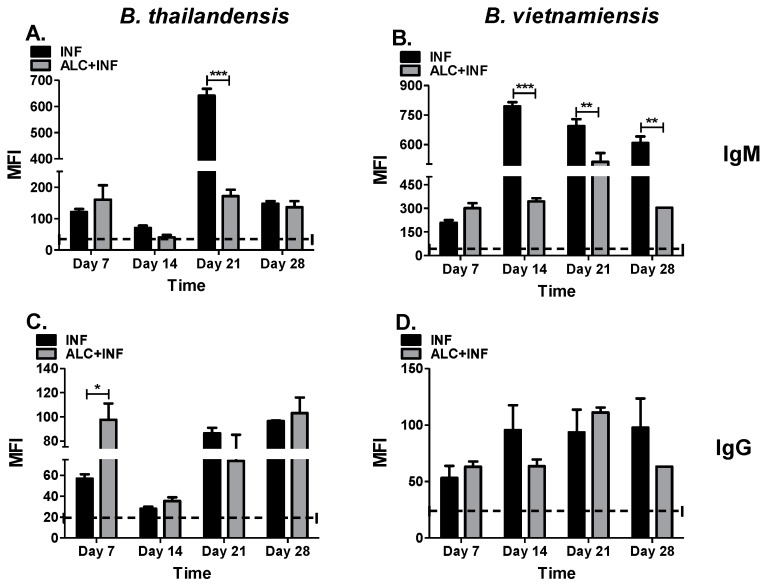
Specific IgM and IgG targeting lipopolysaccharide (LPS)-A of *B. thailandensis* and *B. vietnamiensis*. The mice were treated as described in Figure 3. Whole blood was collected weekly until 28 days PI. LPS-A (5 μg) was coupled to MagPix microspheres and reacted with a 1:1000 serum dilution from infected mice (INF) or mice infected and exposed to alcohol (ALC+INF). Specific LPS-A IgM *B. thailandensis* (**A**) or *B. vietnamiensis* (**B**), or LPS-A IgG *B. thailandensis* (**C**) or *B. vietnamiensis* (**D**) were measured from serum samples. Bars represent mean fluorescence intensity (MFI) per treatment with SEM. Dashed lines represent base values of control groups (CON or ALC). Horizontal line represents statistical comparison between INF and ALC+INF among weekly blood collections determined by Student’s *t*-test, *n* = 3. * *p* < 0.05, ** *p* ≤ 0.01, and *** *p* ≤ 0.001.

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
