# Peer review of "Impact of Binge Alcohol Intoxication on the Humoral Immune Response during Burkholderia spp. Infections"

_microorganisms, 2019, doi:10.3390/microorganisms7050125_

Round 1

Reviewer 1 Report

The manuscript describes a binge alcohol co-infection murine model and its effects on specific IgM and IgG production. Two bacterial agents were used for animal infection studies, B. thailandensis and B. vietnamiensis. Significant differences in mortality and tissue colonisation were observed in alcohol treated B. vietnamiensis infected animals. Differences in IgM and IgM were also observed.

The manuscript is sound and well written. The experiments are well-controlled and the results are interpreted accurately. I only have a few minor queries:

in the introduction and/ or the discussion please provide an overview of the role of the adaptive and humoral immune response in melioidosis.

on line 93, please state what the alcohol was dissolved in.

Sentence beginning 'By day 28 PI....' on line 202 is unclear. It comes across as if you are stating that there is an increase in the control mice but a decrease in the ALC+INF group.

On line 211 you state that alcohol can 'down-modulate' the IgM response. Is it more accurate to state that the rate of the IgM response is delayed in INF+ALC compared to INF group? Would it also be fair to state that the suppression of IgM in B. thailandensis infected animals has a positive effect on clearance of Bth and survival of mice, and that an increase on day 21 in ALC Bv-infected animals has a negative effect on survival?

In Figure 4, why are the levels of total IgG in CON and ALC groups in panel A and B different (4mg/ml and 8mg/ml respectively)?

Figures 5,6 and 7, what are the dashed lines? You state they are base lines in Figure 7. Does this mean limit of detection or something else.

Figure 5, why were different dilutions of serum tested for the IgM and IgG determinations?

Author Response

Dear reviewer,

Thank you for the opportunity to revise our manuscript, Impact of Binge Alcohol Intoxication on the Humoral Immune Response During Burkholderia spp. Infection. We appreciate the careful review and constructive suggestions. The manuscript is substantially improved after making the suggested edits. Our responses are in italics, including how and where the text was modified. Changes made in the manuscript are highlighted via track changes.

Comments: In the introduction and/ or the discussion please provide an overview of the role of the adaptive and humoral immune response in melioidosis.

Response: We have now provided this background in the discussion section on the role of adaptive immunity during melioidosis with two extra references. Lines 339-349.

Healey GD, Elvin SJ, Morton M, Williamson ED. Humoral and cell-mediated adaptive immune responses are required for protection against Burkholderia pseudomallei challenge and bacterial clearance postinfection. Infect Immun. 2005;73:5945–51.

Comments: On line 93, please state what the alcohol was dissolved in.

Response:Yes, clarification regarding the diluent (i.e. PBS) for the 100% alcohol has been included in line 97.

Comments: Sentence beginning 'By day 28 PI....' on line 202 is unclear. It comes across as if you are stating that there is an increase in the control mice but a decrease in the ALC+INF group.

Response:Yes, we agree. The sentence indicating a reduction in IgM levels in the ALC+INF group by 28 days compared to controls has been revised to clarify that the observation is based on reductions in the treatment group and not increases in the control mice, included in lines 211-212.

Comments: On line 211 you state that alcohol can 'down-modulate' the IgM response. Is it more accurate to state that the rate of the IgM response is delayed in INF+ALC compared to INF group? Would it also be fair to state that the suppression of IgM in B. thailandensis infected animals has a positive effect on clearance of Bth and survival of mice, and that an increase on day 21 in ALC Bv-infected animals has a negative effect on survival?

Response:Yes, we concur regarding the clarification of “delayed” vs. “down-modulate” in terms of the rate of the IgM response, as included in line 222. Additionally, it is difficult to accurately indicate a positive or negative relationship of IgM in terms of Burkholderia species infection. The individual IgM responses may have been affected by species specific differences in innate immunity (upstream) that directly or indirectly contributes to a significant IgM response (downstream). An increase in IgM may not have been necessary in ALC+INF (B. thailandensis) mice or alternatively, a positive suppression effect.

Comments: In Figure 4, why are the levels of total IgG in CON and ALC groups in panel A and B different (4mg/ml and 8mg/ml respectively)?

Response: Thank you for the excellent observation. Variations among basal Ig expression in female C57Bl/6 mice are not uncommon. Differences between inter-group dynamics are less useful than intra-group differences across the two bacterial species. In other words, CON and ALC groups remained constant within each bacterial species study.

Comments: Figures 5,6 and 7, what are the dashed lines? You state they are base lines in Figure 7. Does this mean limit of detection or something else?

Response: Thank you for the good question. The dashed lines found on figures 5, 6, and 7 indicate the average expression of IgM or IgG in basal (non-infected) mice across the 28-day study. No statistical differences were determined among weekly Ig levels in non-infected mice. The dashed line serves as a means to determine a signal to noise ratio when examining specific Ig proteins for each respective Burkholderia species.

Comments: Figure 5, why were different dilutions of serum tested for the IgM and IgG determinations?

Response: Different dilutions of sera were used to ensure that values from our samples were within the straight line of the standard curve.

Reviewer 2 Report

Dear Editor,

This manuscript describes the effects of binge alcohol intoxication that potentially suppressed the humoral immune responses in C57BL/6 mice. Burkholderia thailandensis and B. vietnamiensis were used in testing the effects of binge alcohol in this murine model. Most parts of this manuscript are well written. However, additional experiments may be needed to confirm their findings.Here are some comments that need to be addressed.

The authors will need to demonstrate whether the C57BL/6 mouse is a suitable model for B. thailandensis infection. Because there were no mice died or being sick from the infection (in Fig 1A), this has determined that the authors did not use a proper mouse strain and/or infective dose. 

ELISA results will need to be validated by western blot analysis. This can help verify if the isolated CPS or LPS from B. vietnamiensis or B. thailandensis strains were immunological reactive with the mouse sera.

It was not discussed whether the complement system did enhance the ability of antibodies and phagocytic cells to clear B. thailandensis in blood. Is B. thailandensis susceptible to the complement system?

In this study, the binge alcohol intoxication has shown to be involved in the reduction of humoral immunity responses in B. vietnamiensis infection more than that of the infection by B. thailandensis. Please discuss whether binge alcohol would or would not affect to B. pseudomallei infection in your opinion since genetically B. thailandensis is more closely related to B. pseudomallei than B. vietnamiensis.

In the method, it was not mentioned if the mice were anesthetized prior to the intranasal inoculation with the bacteria. It is unusual to inoculate intranasally without anesthetizing the mice.

It is not clear if the CPS was from B. pseudomallei or B. vietnamiensis used in testing the IgM or IgG responses shown in Fig 6. Please provide the method the authors used to prepare the CPS from B. vietnamiensis if the CPS was extracted from B. vietnamiensis in this study. 

Because the authors used ELISA and MagPix assays in this study, it was mentioned in the figure description the method used to measure the IgM or IgG. 

Author Response

Dear reviewer,

Thank you for the opportunity to revise our manuscript, Impact of Binge Alcohol Intoxication on the Humoral Immune Response During Burkholderia spp. Infection. We appreciate the careful review and constructive suggestions. The manuscript is substantially improved after making the suggested edits. Our responses are in italics, including how and where the text was modified. Changes made in the manuscript are highlighted via track changes.

Comments: The authors will need to demonstrate whether the C57BL/6 mouse is a suitable model for B. thailandensis infection. Because there were no mice died or being sick from the infection (in Fig 1A), this has determined that the authors did not use a proper mouse strain and/or infective dose.

Response: We disagree with Reviewer 2 that we had selected the wrong strain of mice because no mortality was observed. Our goal was not to observe mortality but to determine if alcohol had a negative effect on the host immune system during infection with a low virulence strain (B. thailandensis strain). In nature both, B. thailandensis and the virulent B. pseudomallei are found in soil and likely co-infect patients.

Current animal models of Burkholderia infection focus primarily on acute disease. For example, rats and Syrian hamsters have been shown to be susceptible to infection, which results in acute disease and rapid death. In mouse models of Burkholderia infection, BALB/c mice are very susceptible, whereas C57BL/6 mice are relatively more resistant via intravenous and intranasal routes of infection. Mortality in BALB/c mice is due to septicemic disease with overwhelming bacterial loads in organs and blood.

Leakey AK, Ulett GC, Hirst RG. BALB/c and C57Bl/6 mice infected with virulent Burkholderia pseudomallei provide contrasting animal models for the acute and chronic forms of human melioidosis, Microb Pathog, 1998, vol. 24(pg. 269-275).

Hoppe I, Brenneke B, Rohde M, Kreft A, Häußler S, Reganzerowski A, Steinmetz, I. Characterization of a murine model of melioidosis: comparison of different strains of mice, Infect Immun, 1999, vol. 67 (pg. 2891-2900).

Comments: ELISA results will need to be validated by western blot analysis. This can help verify if the isolated CPS or LPS from B. vietnamiensis or B. thailandensis strains were immunological reactive with the mouse sera.

Response: Our intent was not to identify immunogenic antigens during infection but the overall humoral response to infection and the impact that alcohol may have during this host-pathogen interaction. Indeed, we appreciate the suggestion from Reviewer 2 but we feel that it is beyond the scope of the present goal. Furthermore, both immunoassays quantify host antibody responses to intact (ELISA) and denatured (Western blot) microbial antigens.

Comments: It was not discussed whether the complement system did enhance the ability of antibodies and phagocytic cells to clear B. thailandensis in blood. Is B. thailandensis susceptible to the complement system?

Response: Thank you for the excellent question and observation. Although outside the scope of our study, B. thailandensis is susceptible to the complement system, however compliment dysfunction under single binge alcohol conditions have not been examined. Additional discussion has been added to address potential differences in complement and innate immunity between acapsular B. thailandensis and capsular producing B. vietnamiensis that may explain, at least in part, bacterial clearance with B. thailandensis in lines 368-370.

Woodman ME, Worth RG, Wooten RM (2012) Capsule Influences the Deposition of Critical Complement C3 Levels Required for the Killing of Burkholderia pseudomallei via NADPH-Oxidase Induction by Human Neutrophils. PLoS ONE 7(12): e52276. doi:10.1371/journal.pone.0052276

Comments: In this study, the binge alcohol intoxication has shown to be involved in the reduction of humoral immunity responses in B. vietnamiensis infection more than that of the infection by B. thailandensis. Please discuss whether binge alcohol would or would not affect to B. pseudomallei infection in your opinion since genetically B. thailandensis is more closely related to B. pseudomallei than B. vietnamiensis.

Response: Thank you for the comment. Although a more severe effect was observed during a B. vietnamiensis infection and binge alcohol intoxication, generated results indicate a binge alcohol mediated dysfunction in humoral immunity for both B. pseudomallei near-neighbor species provides valuable insight for potential effects during a B. pseudomallei infection.  As discussed in the introduction, both species provide a more complete surrogate system to understand binge alcohol effects when considering B. pseudomallei virulence and intracellular host survival. We have included our prediction and brief commentary based on the findings of the current study regarding the effect of binge alcohol intoxication during a B. pseudomallei infection in lines 403-406.

Comments: In the method, it was not mentioned if the mice were anesthetized prior to the intranasal inoculation with the bacteria. It is unusual to inoculate intranasally without anesthetizing the mice.

Response: Yes, we agree. The mice were intranasally inoculated under “light anesthesia” conditions (i.e. isoflurane). No adverse or additive effects were measured as a result of the anesthesia. A description of anesthesia during the intranasal inoculation methods have been included to improve the clarity of the methods in lines 100-101.

Comments: It is not clear if the CPS was from B. pseudomallei or B. vietnamiensis used in testing the IgM or IgG responses shown in Fig 6. Please provide the method the authors used to prepare the CPS from B. vietnamiensis if the CPS was extracted from B. vietnamiensis in this study.

Response: No, we did not purify CPS from B. vietnamiensis. We used only CPS from B. pseudomallei kindly provided to us by Dr. Paul Brett. He has shown that CPS from B. pseudomallei cross reacts with CPS from B. vietnamiensis. We have clarified this in the text in lines 286 and 296.

Round 2

Reviewer 2 Report

The revised manuscript has some improvements. However, some limitations are still present.

The CPS antigen used in this study was made from B. pseudomallei, not from B. vietnamiensis. Since this B. pseudomallei - CPS antigen is not known to be expressed by B. vietnamensis, using this antigen to determine the specific IgM and IgG responses in sera from mice infected by B. vietnamiensis is invalid. This part of the study should be removed from the manuscript.

The authors did not validate their ELISA and MagPix results by western blotting analysis per the reviewer 2's suggestion. This is a limitation of this study. Western blot can also tell whether the CPS from B. pseudomallei is cross-reactive with the humoral response from B. vietnameinsis infection. Without this validation, this specific humoral response against B. pseudomallei CPS by MagPix cannot be evaluated.

Author Response

We value reviewer 2 comments. However, to have a positive association between the CPS from B. pseudomallei cross reacting with the CPS from B. vietnamiensis a Western blot is the way to go. Unfortunately, our monoclonal antibody obtained from Dr. David Aucoin (3C5) has been shown not to cross react with B. vietnamiensis (Houghton, R.L., et al. 2014. PLOS Neglected tropical Diseases, 8(3):e2727). However, to confirm this cross reactivity other monoclonal antibodies produced by Dr. Aucoin would need to be tested. Regardless, we have indirect evidence that the CPS from B. pseudomallei cross reacts with CPS from B. vietnamiensis as shown by the increased absorbance values in both the ELISA and MagPix assays. See lines 386 to 394.